# [Re] Smoothed Energy Guidance for Diffusion Models

# ML Reproducibility Challenge-2025

## Abstract

This study is part of the MLRC Reproducibility Challenge 2025, aiming to reproduce and improve the results from a NeurIPS 2024 submission *Smoothed Energy Guidance (SEG): Guiding Diffusion Models with Reduced Energy Curvature of Attention.* The work proposed in the SEG paper faced key limitations, including the lack of an ablation study for optimal kernel size selection and unexplored alternative blurring strategies within diffusion models, which could offer valuable insights into enhancing image quality and model robustness. Furthermore, the approach employed unnecessary smoothing throughout all iterations of the denoising process, which not only diminished the clarity of the output but also resulted in increased computational costs. To address these issues, we conducted a detailed ablation study and explored more efficient alternatives, including Exponential Moving Average (EMA) and BoxBlur using integral images, to improve computational efficiency while maintaining image quality. Our findings provide insights into optimizing smooth energy guidance in diffusion models, reducing computational overhead while improving image quality. Our code is available at SEG Reproducibility Challenge 2025 Repository.

## 1 Introduction

Smoothed Energy Guidance (SEG) was proposed as a tuning- and condition-free method for improving self-attention mechanisms from an energy-based perspective. Instead of relying on a guidance scale parameter, SEG directly blurs attention weights using a Gaussian kernel Gonzalez-Jimenez et al. (2021b) to reduce the curvature of the underlying energy function. This approach enables continuous control over energy modulation while avoiding the side effects of large guidance scales. A novel query blurring technique was proposed in the SEG paper to achieve full attention weight smoothing without quadratic cost using Eq.1.

$$\text{SEG}(\mathbf{Q}, \mathbf{K}, \mathbf{V}) = \text{softmax}\left(\frac{\mathcal{B}(\mathbf{Q})\mathbf{K}^T}{\sqrt{d}}\right)\mathbf{V} \tag{1}$$

Where $\mathcal{B}$ is the Toeplitz matrix for Gaussian blur. $\mathbf{Q}$ has shape $(B, H, T, E)$, where $B$ is the batch size, $H$ is the number of attention heads, $T$ is the number of tokens, and $E$ is the embedding dimension.

Furthermore, the SEG paper revised the Stochastic Differential Equation (SDE) for Diffusion Models by incorporating energy guidance, which leads to improved image generation quality Nichol & Dhariwal (2021). Eq. 2 is the updated formulation:

$$d\mathbf{x} = \left[\mathbf{f}(\mathbf{x}, t) - g(t)^2\left((1 - \gamma_{\text{cfg}} + \gamma_{\text{seg}})s_\theta(\mathbf{x}, t) + \gamma_{\text{cfg}}s_\theta(\mathbf{x}, t, c) - \gamma_{\text{seg}}\tilde{s}_\theta(\mathbf{x}, t)\right)\right]dt + g(t)d\bar{\mathbf{w}} \tag{2}$$

where $s_\theta$ is the score model and $\tilde{s}_\theta$ is the score model calculated using blurred attention weights. The parameters $\gamma_{\text{cfg}}$ and $\gamma_{\text{seg}}$ control the guidance strength of classifier-free guidance Ho & Salimans (2022b) and energy-based guidance, respectively Wang (2023).

## 1.1 Scope of Reproducibility

This investigation focuses on several critical aspects concerning the reproducibility of the SEG paper:

- **Analysis of Kernel Size Variations in Energy-Guided Generation**: The influence of energy guidance on the quality of image generation, particularly regarding the application of varying Gaussian blur kernel sizes, has not been thoroughly analyzed or discussed.

- **Impact of Gaussian Blur on Different Attention Layers in U-Net**: There is ambiguity regarding the specific set of attention layers (down, mid, up) within the U-Net architecture, to which part of the U-Net Gaussian blur should be applied, aiming to smooth the energy landscape and enhance the quality of image generation.

- **Minimizing Redundant Computations in Blurring Iteratively**: The application of blurring at each iteration to smooth the energy landscape results in unnecessary computations that could be eliminated to improve efficiency.

## 2 Related Work

Diffusion models Ho et al. (2020); Song et al. (2020) represent a powerful class of generative models, excelling in tasks like image synthesis. They operate via a forward noising process and a learned reverse denoising process, often implemented with U-Net architectures Ronneberger et al. (2015); Rombach et al. (2022a). Enhancing the quality and controllability of generated samples often involves guidance mechanisms.

### 2.1 Guidance Mechanisms in Diffusion Models

**Classifier-Free Guidance (CFG) :**  A dominant technique for conditional generation is CFG Ho & Salimans (2022a). It steers the sampling process by interpolating between conditional and unconditional model predictions, controlled by a guidance scale $\gamma_{cfg}$. While effective in improving prompt alignment and sample fidelity, CFG necessitates evaluating the model twice per step (conditionally and unconditionally), increasing computational cost, and high guidance scales can sometimes impair sample diversity Sadat et al. (2024).

**Smoothed Energy Guidance:**  Proposed by Hong et al. Hong et al. (2024), SEG offers a training-free and condition-free guidance approach derived from an energy-based view of the self-attention mechanism, linking it to modern Hopfield networks Ramsauer et al. (2021). It modulates the generation by blurring attention weights (or queries, for efficiency) using a Gaussian kernel Gonzalez-Jimenez et al. (2021a), with the blur strength $\sigma$ acting as the control parameter. This aims to smooth the energy landscape and improve quality without the drawbacks of high CFG scales. The original work proposed a modified SDE incorporating both CFG ($\gamma_{cfg}$) and SEG ($\gamma_{seg}$) terms (formulated in Eq 2). Key aspects related to SEG include the impact of kernel size variations, the effect of blurring different U-Net attention layers, the potential use of alternative blurring strategies, and optimizing the temporal application interval during inference.

**Perturbation-Based Guidance :**  SEG falls under perturbation-based guidance methods, which use altered model states or inputs to steer generation. Karras et al. Karras et al. (2024) proposed Autoguidance, using a weaker model variant to improve quality while preserving diversity. Self-Attention Guidance (SAG) Hong et al. (2022) blurs attention-based regions, while Perturbed-Attention Guidance (PAG) Ahn et al. (2024) perturbs attention maps (e.g., using identity matrices) to guide sampling. While effective, these methods rely on heuristic perturbations, often causing artifacts like smoothed details, color shifts, and structural changes under strong guidance. Additionally, their theoretical foundations are not well established. Like SEG, they operate without external classifiers, instead leveraging internal signals. Our work extends this line by analyzing various perturbation types (Gaussian, EMA, Interpolated BoxBlur) within the SEG framework.

## 2.2 Optimizing Guidance Application

A key finding of our study is that applying SEG blurring only during the initial stages (see section 4.3) of the reverse diffusion process is sufficient for guidance, significantly reducing computational load without sacrificing output quality. This observation resonates with recent findings in the CFG literature suggesting that constant guidance application throughout inference is suboptimal. Kynkäänniemi et al. Kynkäänniemi et al. (2024) demonstrated that CFG yields the most benefit within a specific interval of noise levels (often mid-process), being potentially harmful early on and unnecessary late. Restricting CFG to such an interval improved results and efficiency. Castillo et al. Castillo et al. (2023) developed Adaptive Guidance to dynamically omit CFG calculations when updates converge, reducing NFEs. Furthermore, research into CFG weight schedulers Wang et al. (2024) and condition annealing Sadat et al. (2024) explores dynamically adjusting guidance strength over time. These studies collectively support the principle of non-constant or temporally limited guidance application, aligning with our findings for SEG.

## 2.3 Computational Efficiency of Guidance

While the overall computational cost of diffusion models is high due to their iterative nature, the choice of guidance mechanism also contributes. CFG inherently doubles the computation per step Ho & Salimans (2022a). SEG avoids this doubling but introduces its own cost via the blurring operation. The original Gaussian blurring has a complexity related to the kernel size (e.g., $O(T \times k^2)$) Table 2). Our work addresses this by showing that blurring is not needed in all steps and by proposing computationally cheaper alternatives like EMA and Interpolated BoxBlur (via integral images), making SEG more efficient.

## 3 Experimental Setup and Code

**What was easy**: The codebase provided by the authors of the SEG paper is well-documented, making it easy to conduct experiments with varying blur strength ($\sigma$). To enhance clarity and maintainability, we have modularized the codebase into distinct components, separating key conceptual elements such as the model pipeline, blurring strategies, and attention_processor modules.

**What was difficult**: The original code was constrained to studying the effects of Gaussian blurring with varying $\sigma$ and guidance strength ($\gamma_{\text{seg}}$), limiting flexibility in experimentation. To enhance the reproducibility and efficiency of the code, we optimized the 2D Gaussian blur convolution by caching the Gaussian kernel, reducing computational overhead. Additionally, we introduced flexible hyperparameter settings, allowing adjustments to $\sigma$, kernel size, blurring schedules, and selective blurring of attention layers. Furthermore, we expanded the model pipeline to support alternative, computationally efficient smoothing strategies beyond Gaussian convolution, enabling broader exploration of image generation quality improvements. To quantitatively assess the impact of blurring, we incorporated various metrics like Frobenius Norm Böttcher & Wenzel (2008), Laplacian Variance Bansal et al. (2016), and Gradient Entropy Zhao et al. (2016) applied to attention layers of the mid-block of the U-Net of the diffusion model Rombach et al. (2022b).

## 4 Experiments

The key parameters employed for SEG-based generation are summarized in Table 1. Further, Figure 1 illustrates the variation in FID scores for unconditional generation across different values of $\sigma$ and $\gamma_{\text{seg}}$. As observed, increasing $\gamma_{\text{seg}}$ generally leads to improved FID scores, with diminishing returns beyond $\gamma_{\text{seg}} = 3$. Based on this trend, we adopt a fixed setting of $\gamma_{\text{seg}} = 3$ for all subsequent experiments to balance performance gains.

### 4.1 Analysis of Kernel Size Variations in Energy-Guided Generation

The SEG paper defines the kernel size as a function of $\sigma$ using the following relation:

$$\text{kernel size} = \lceil 6\sigma \rceil + 1 - (\lceil 6\sigma \rceil \bmod 2) \tag{3}$$

| Parameter | Value |
|---|---|
| Blur Time | [begin] |
| $\gamma_{\text{cfg}}$ | 0 (unconditional generation) |
| num_inference_steps | 30 |
| SEG applied layers | [mid] |
| $\gamma_{\text{seg}}$ | 3 |
| width | 1024 |
| height | 1024 |
| Diffusion Model | stabilityai/stable-diffusion-xl-base-1.0 (Pretrained) |

Table 1: Parameters used for SEG reproducibility. **Blur Time**: The inference steps were divided into three parts, [begin] means SEG was applied to the initial 1/3 of the inference steps. **SEG applied layers**: The attention layers of U-Net to which SEG (query blurring) was applied. These are the default parameters used for SEG analysis unless explicitly stated.

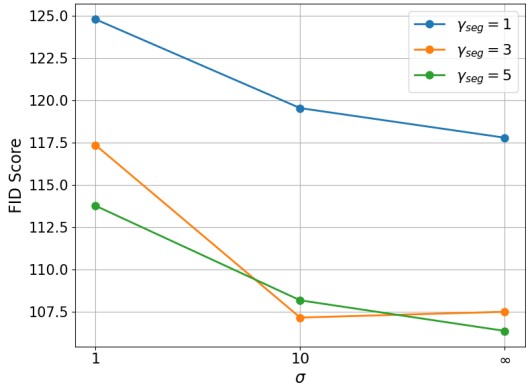

Figure 1: Variation of FID for unconditional generation under varying Gaussian blur strength ($\sigma$) and $\gamma_{\text{seg}}$ with kernel size derived using Eq 3 for Gaussian based SEG.

For $\sigma = 10$, Eq 3 yields a kernel size of 61, which is computationally impractical. This is because the blurring is applied along the token dimension $T$ of the query tensor $\mathbf{Q}$, which has the shape $(B, H, T, E)$. The operation is performed independently for each combination of batch $B$, attention head $H$, and embedding dimension $E$. Here, $T$ corresponds to a flattened latent feature map, typically of size 1024 ($32 \times 32$) or 4096 ($64 \times 64$). Although the blur is applied in latent space, convolving over such a long sequence with a large kernel size in each dimension leads to significant computational overhead.

Table 2 and Figure 5 illustrate the computational burden of large kernel sizes. To understand the trade-off between image quality and efficiency, we conducted a series of experiments varying the kernel size. Figure 2 shows images generated using different kernel sizes. Along with visual inspection, quantitative analysis of this trade-off is captured in Table 3.

### 4.2 Impact of Gaussian Blur on Different Attention Layers in U-Net

The U-Net part of the latent diffusion models typically includes multiple attention layers in their down, mid, and up regions. We systematically evaluated the impact of applying Gaussian blur to each region, which has not been reported in the SEG paper. The effect of blurring at different attention layers of U-Net is shown in Figure 3.

- **Down and UP Attention Layers**: Blurring at the down and up regions of the U-Net resulted in highly distorted and unnatural images. The color and texture appeared scrambled, with a significant

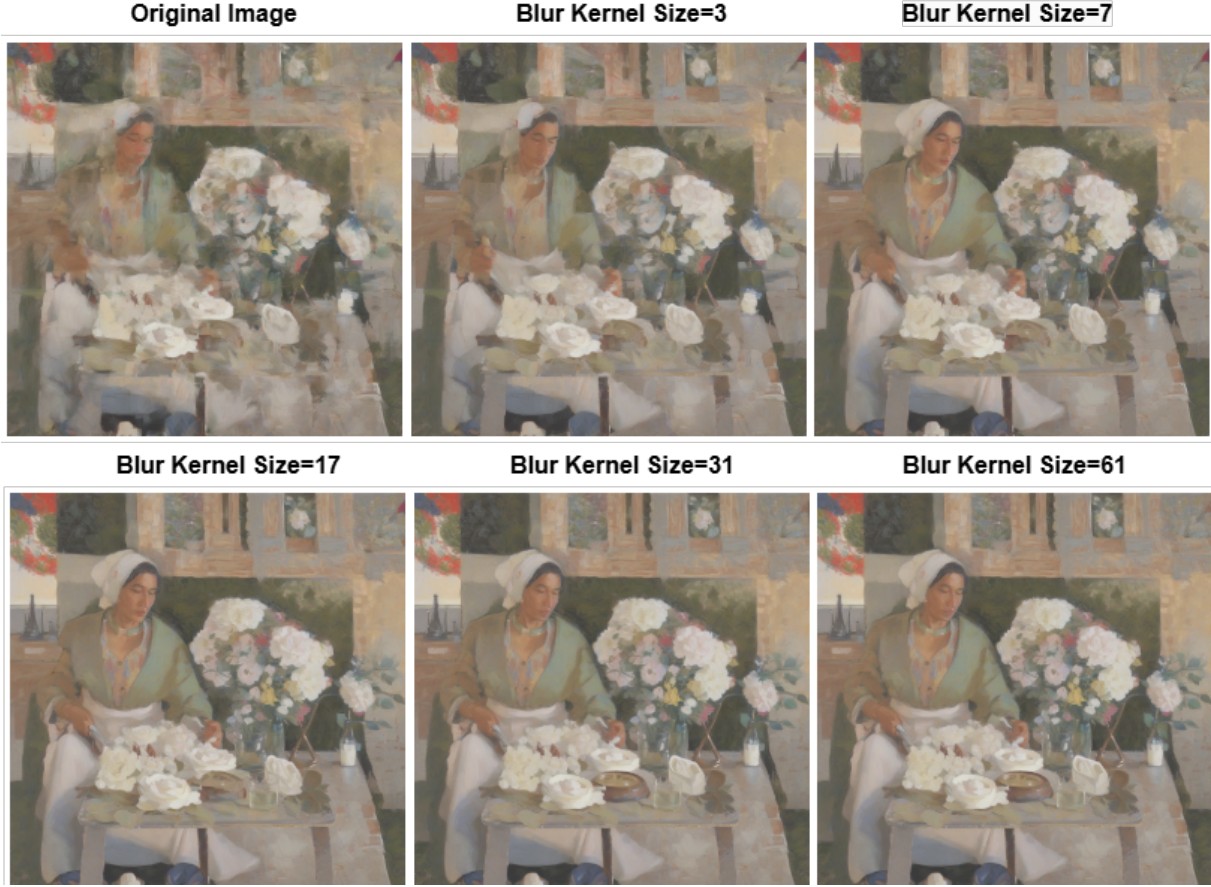

Figure 2: Image (unconditional, seed=77) quality assessment using Gaussian blur with varying kernel sizes, where $\sigma$=10. More seeds are included in Appendix (Figure 10 and 11)

loss of structure and coherence. Instead of smooth transitions or meaningful abstractions, the images exhibited disorganized patterns and blocky artifacts.

- **Mid Attention Layers**: The middle images in the second row, which appear more natural than the others, suggest that blurring in the mid-layer impacts the overall structure without completely disrupting image coherence. While textures are slightly smoothed, the images still preserve recognizable forms and objects.

The observations from the images confirm that selective blurring affects different regions of U-Net differently. Applying it at the mid-region is the most balanced approach while blurring in the down or up regions severely degrades image quality. The authors of the original SEG paper did not discuss their strategy of which attention layers to apply SEG to.

### 4.3 Minimizing Redundant Computations in Blurring Iteratively

Applying blurring at each iteration to smooth the energy landscape leads to unnecessary computations, which can be optimized for improved efficiency. Figure 3 illustrates the image quality achieved when blurring is applied at different stages of the iteration process (different Blur Time).

- In the SEG paper, Gaussian blurring was applied at every iteration, significantly increasing computational costs. However, as obvious from Figure 3, Blur Time= Begin is able to retain the global

structure of the image, revealing that applying Gaussian blur only during the first 1/3 of the reverse diffusion steps is sufficient to guide the model toward stable solutions.

It is important to recognize that while smoothening attenuates sharp variations in the energy landscape—thereby stabilizing the diffusion process—it must be applied judiciously. Excessive smoothening, particularly in the later stages of reverse diffusion, can lead to a significant loss of fine-grained information and semantic detail, ultimately resulting in a distorted generation trajectory. This highlights a key trade-off in the design of smoothening schedules: early smoothening tends to be more effective, as the initial diffusion steps operate on highly noisy latent where structural information is sparse and regularization is most impactful. In contrast, late-stage smoothening risks oversuppressing emerging image subtleties. Thus, controlling the timing and intensity of smoothening is critical to ensure stability without compromising generative fidelity.

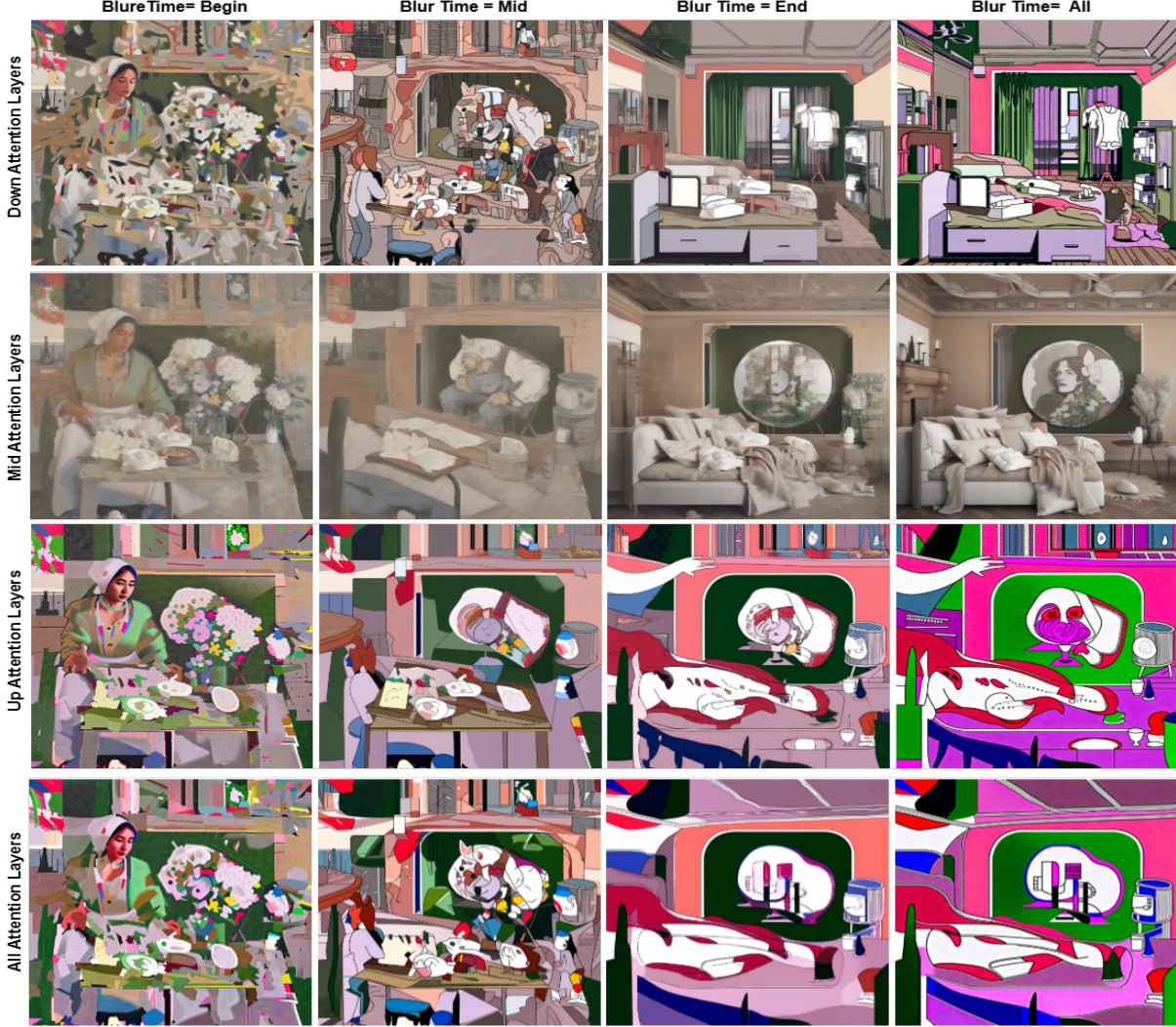

Figure 3: Image (unconditional, seed=77) quality generation for a fixed kernel size of 31 and $\sigma = 10$. The 30 iterations of reverse diffusion are divided into three equal phases: Begin (iterations 1-10), Mid (iterations 11-20), and End (iterations 21-30). More generated examples are provided in Appendix (Figures 12 and 13)

## 5 Experiment Beyond Paper

### 5.1 Generalizing the Smoothed Energy Guidance Mechanism

The Smoothed Energy Guidance (SEG) framework Hong et al. (2024) enhances diffusion model sampling by utilizing predictions derived from a smoothed version of the attention mechanism's internal representations. Specifically, it computes a guidance term based on the difference between the standard prediction $s_\theta(x, t)$ and a prediction $\tilde{s}_\theta(x, t)$ obtained after applying a blur operation to the query tensor $Q$ (equivalent to blurring the attention map $A = QK^T$). The core theoretical motivation is that this blurring attenuates the curvature of the energy landscape $E(A)$ associated with attention. While the original work focuses on 2D Gaussian blur, the underlying principle can be generalized by identifying the essential properties required of the blurring operation.

Let the blurring process be represented by a linear operator (matrix) $B$ acting on the vectorized attention map $a$. The key requirements for $B$ to be suitable within the SEG framework, ensuring the smoothed prediction $\tilde{a} = Ba$ corresponds to an attenuated energy landscape, are:

1. **Mean Preservation:** The operation must preserve the mean value of the input signal. If $\mu = \mathbb{E}[a]$, then the expected value of the blurred signal should remain $\mathbb{E}[\tilde{a}] = \mathbb{E}[Ba] = \mu$. This ensures the overall signal level is maintained, as noted for Gaussian blur in (Hong et al., 2024, Lemma 3.1).

2. **Variance Reduction:** The operation must smooth the input, necessarily reducing its variance. That is $Var[\tilde{a}] = Var[Ba] \leq Var[a]$, with strict inequality for non-trivial smoothing. This property is fundamental to smoothing and is also demonstrated for Gaussian blur in (Hong et al., 2024, Lemma 3.1).

3. **Kernel Weight Constraint:** When the blur is represented by the matrix $B$, its diagonal elements $b_{ii}$ (representing the influence of an element $a_i$ on its blurred counterpart $\tilde{a}_i$) must satisfy $0 \leq b_{ii} < 1$. This reflects the averaging nature of blurring, ensuring no single point fully determines its blurred value. This condition is crucial for the Hessian determinant analysis presented in (Hong et al., 2024, Theorem 3.1, Appendix A.3) which aims to show $|det(\tilde{H})| < |det(H)|$. The approximation $|det(\tilde{H})| \approx \prod_i \xi(\tilde{a})_i b_{ii}$ directly relies on $b_{ii} < 1$.

The variance reduction property (2) is directly linked to the smoothing of the energy landscape. Using the second-order Taylor approximation around the mean $\mu$ as in (Hong et al., 2024, Lemma 3.2, Appendix A.2), the term $\sum_{ij} e^{a_{ij}}$ within the energy function's log-sum-exp (*lse*) term can be approximated as being proportional to $(1 + \frac{1}{2} Var[a])$. Therefore, any blur operator $B$ satisfying properties (1) and (2) will reduce $Var[a]$, decrease the approximate value of $\sum_{ij} e^{\tilde{a}_{ij}}$, and thus increase the approximate $lse(\tilde{a})$. This increase in the *lse* term corresponds to a shift towards uniformity, indicative of a smoother energy landscape with attenuated curvature.

#### 5.1.1 A Momentum-Based Interpretation of Energy Guidance

In this section, we interpret the SDE equation of diffusion models through the lens of deep learning optimizers. We provide a mathematically equivalent perspective on the linear combination of different score models $\mathbf{s}_\theta$ as described in Eq. 2.

$$dx = \left[ \mathbf{f}(\mathbf{x}, t) - g(t)^2 \mathbf{s}_\theta^\star(\mathbf{x}, t) \right] dt + g(t) d\bar{\mathbf{w}} \tag{4}$$

where

$$\mathbf{s}_\theta^\star(\mathbf{x}, t) = (1 - \gamma_{\text{cfg}} + \gamma_{\text{seg}}) \mathbf{s}_\theta(\mathbf{x}, t) + \gamma_{\text{cfg}} \mathbf{s}_\theta(\mathbf{x}, t, c) - \gamma_{\text{seg}} \tilde{\mathbf{s}}_\theta(\mathbf{x}, t) \tag{5}$$

$$= \mathbf{s}_\theta(\mathbf{x}, t) + \gamma_{\text{cfg}}(\mathbf{s}_\theta(\mathbf{x}, t, c) - \mathbf{s}_\theta(\mathbf{x}, t)) + \gamma_{\text{seg}}(\mathbf{s}_\theta(\mathbf{x}, t) - \tilde{\mathbf{s}}_\theta(\mathbf{x}, t)) \tag{6}$$

In Eq. 6, $\mathbf{s}_\theta(\mathbf{x}, t)$ represents the unguided score model. The term $(\mathbf{s}_\theta(\mathbf{x}, t, c) - \mathbf{s}_\theta(\mathbf{x}, t))$ accounts for classifier guidance, while $(\tilde{\mathbf{s}}_\theta(\mathbf{x}, t) - \mathbf{s}_\theta(\mathbf{x}, t, c))$ represents energy guidance. The final resultant score model is denoted as $\mathbf{s}_\theta^\star(\mathbf{x}, t)$.

Assuming $\gamma_{\text{cfg}} = 0$, we obtain:

$$\mathbf{s}_\theta^\star(\mathbf{x}, t) = \mathbf{s}_\theta(\mathbf{x}, t) + \gamma_{\text{seg}}(\tilde{\mathbf{s}}_\theta(\mathbf{x}, t) - \mathbf{s}_\theta(\mathbf{x}, t, c)) \tag{7}$$

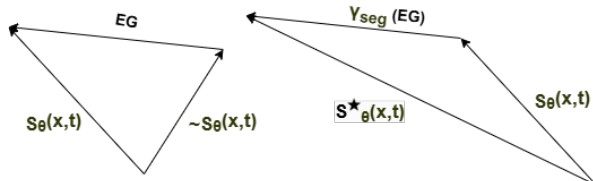

Figure 4: Illustration of energy guidance in diffusion models, drawing an analogy to momentum-based optimizers like Adam, where $\gamma_{\text{seg}}$ influences the update direction similar to accumulated momentum, leading to a smoother optimization landscape. EG stands for Energy Guidance.

Figure 4 suggests that energy guidance functions similarly to momentum in optimizers like Adam, where the final gradient incorporates both the current gradient and accumulated momentum, leading to a smoother loss landscape. As demonstrated in the SEG paper, generation quality improves for extreme blur strength $(\sigma \to \infty)$. In this scenario, maximal information destruction occurs, shifting from weighted averaging via a Gaussian kernel to simple averaging, which increases the orthogonality between $\mathbf{s}_\theta$ and $\tilde{\mathbf{s}}_\theta$. This enhanced orthogonality plays a key role in improving generation quality.

The constraints required to satisfy the SEG framework, as outlined in Section 5.1, together with the orthogonality perspective discussed above, pave the way for investigating alternative smoothening techniques that are both well-suited to the framework and computationally more efficient. In the following section, we explore two such techniques in detail.

| Blurring Technique | Time Complexity | Optimal Parameter Ranges |
|---|---|---|
| Gaussian Blur | $O(T \times k^2)$ | k = 31, $\sigma = 10$ |
| EMA Smoothing | $O(T)$ | $\beta = 0.95$ |
| Box Blur | $O(T)$ | $\alpha = 0.99$ |

Table 2: Comparison of Blurring Techniques: Time Complexity and Optimal Parameter Ranges. Here, T represents the token count in a query with the shape (batch, attention head, token, embedding dimension), while K denotes the kernel size

### 5.1.2 Exponential Moving Average Smoothing

Exponential Moving Average (EMA) smoothing is a widely used technique for smoothing sequential data, where the influence of past observations decays exponentially over time. It is commonly applied in time series analysis and has found extensive use in deep learning tasks. The recursive formulation of EMA is given by:

$$\text{EMA}_t = \beta x_t + (1 - \beta)\text{EMA}_{t-1} \tag{8}$$

Here, $x_t$ denotes the token at position $t$ along the token dimension $T$ (typically $T = 32 \times 32$ or $64 \times 64$), and $\beta \in (0, 1)$ is the smoothing factor. A larger $\beta$ places more emphasis on the current token value (less smoothing), while a smaller $\beta$ gives greater weight to the accumulated past, resulting in a smoother output.

In this work, we adapt EMA smoothing for application to the query tensor $Q$. Our implementation uses a parameterization equivalent to setting $\beta$ close to 1 (e.g., $\beta = 0.85$ or $\beta = 0.95$), emphasizing recent

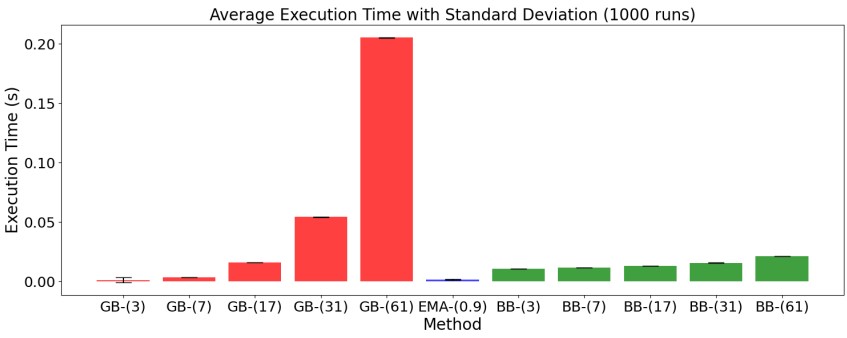

Figure 5: Quantitative speed comparison of blurring techniques (GB-$i$: Gaussian Blur with kernel size $i$; EMA-$i$: EMA with smoothing factor $\alpha = i$; BB-$i$: Box Blur with kernel size $i$). EMA and Box Blur provide cheaper alternatives to the computationally intensive Gaussian Blur. The input used for analysis is of shape : [Batch=3, Attn Heads=20, Tokens= 4096 (64x64), Embedding= 64] and dtype: float64

.

information while still incorporating past context with exponentially decaying weights. The theoretical alignment of this specific EMA implementation with the principles of SEG is discussed in Appendix A.1.2.

### 5.1.3 Interpolated Box Blur

Box Blur, also known as a mean filter, is a fundamental smoothing technique where each pixel (or feature) in an input tensor is replaced by the average value of its surrounding neighborhood within a defined kernel. For a given input tensor $I$ and a kernel of size $k \times k$, the standard Box Blur output $I'$ at location $(x, y)$ is computed as:

$$I'_{(x,y)} = \frac{1}{k^2} \sum_{i=-\lfloor k/2 \rfloor}^{\lfloor k/2 \rfloor} \sum_{j=-\lfloor k/2 \rfloor}^{\lfloor k/2 \rfloor} I(x+i, y+j) \tag{9}$$

where $k$ is the kernel size (typically odd). This operation can be efficiently implemented using integral images.

In our work, we employ an *Interpolated Box Blur* approach, which linearly interpolates between the original input tensor $I$ and its box-blurred version $I'$, aiming to achieve significant smoothing while subtly emphasizing the central element compared to a standard mean filter:

$$Img'(x, y) = (1 - \alpha) \cdot I(x, y) + \alpha \cdot I'(x, y) \tag{10}$$

where $\alpha$ is an interpolation coefficient set close to 1 (e.g., $\alpha = 0.9$) to simulate Gaussian with high $\sigma$. The theoretical alignment of this method with the principles of SEG is discussed in Appendix A.1.1.

### 5.2 Quantitative Assessment Through Metrics

This section examines the dynamics of the reverse diffusion process and the impact of SEG smoothening, analyzing trajectories via metrics applied to smoothed hidden states (the latent). Initially an ablation study, the surprising results warranted this focused discussion, which is distinct from exploring new smoothening techniques or validating original SEG claims. Below is a short description of various metrics used for this study.

**Frobenius Norm**: The Frobenius norm provides a scalar value representing the "energy" of a matrix, analogous to the Euclidean norm for vectors. For a matrix $A$ of size $m \times n$, the Frobenius norm is defined as:

$$\|A\|_F = \sqrt{\sum_{i=1}^{m}\sum_{j=1}^{n}|a_{ij}|^2} \tag{11}$$

where $a_{ij}$ represents the element at the $i$th row and $j$th column of matrix $A$.

A higher Frobenius norm indicates that the matrix has larger-valued elements, often implying greater variance, higher energy, or stronger signal intensity. Conversely, a lower Frobenius norm suggests smaller-valued elements, indicating lower energy, reduced variability, or smoother transitions within the matrix structure.

**Laplacian Variance**: A measure of image sharpness, where higher values indicate sharper images and lower values suggest smoothed images. It is computed using the Laplacian operator, which highlights edges through convolution with the 3×3 kernel:

$$K_{\text{Laplacian}} = \begin{bmatrix} 0 & 1 & 0 \\ 1 & -4 & 1 \\ 0 & 1 & 0 \end{bmatrix}$$

For an image $I$, the Laplacian $L(I)$ is obtained by convolving $I$ with $K_{\text{Laplacian}}$. The variance of $L(I)$ is given by:

$$\text{Variance} = \frac{1}{N}\sum_{i=1}^{N}(L(I)_i - \mu)^2 \tag{12}$$

where $L(I)_i$ are Laplacian pixel values, $\mu$ is their mean, and $N$ is the total pixels. Higher variance indicates sharper details.

**Gradient Entropy**: Gradient entropy measures the randomness in image gradients, reflecting texture complexity. Higher entropy indicates more detailed structures, while lower entropy suggests smoother regions.

- **Gradient Calculation**: Compute horizontal and vertical gradients using Sobel operators:

$$G_x = I * K_x, \quad G_y = I * K_y \tag{13}$$

  where $I$ represents the input image, $K_x$ and $K_y$ are the Sobel kernels for computing horizontal and vertical gradients, respectively, and $*$ denotes the convolution operation.

- **Gradient Magnitude**: Calculate the overall gradient magnitude:

$$G = \sqrt{G_x^2 + G_y^2} \tag{14}$$

  where $G_x$ and $G_y$ represent the horizontal and vertical gradients.

- **Entropy Formula**: Measure the randomness of the gradient magnitudes:

$$H(G) = -\sum_i p_i \log_2(p_i) \tag{15}$$

  where $H(G)$ denotes the gradient entropy, $p_i$ represents the probability of the $i^{th}$ gradient magnitude value, and $\log_2$ refers to the logarithm to base 2.

## 6 Results and Discussion

This section explored the impact of different blurring techniques, including EMA and Box blur, on the image quality.

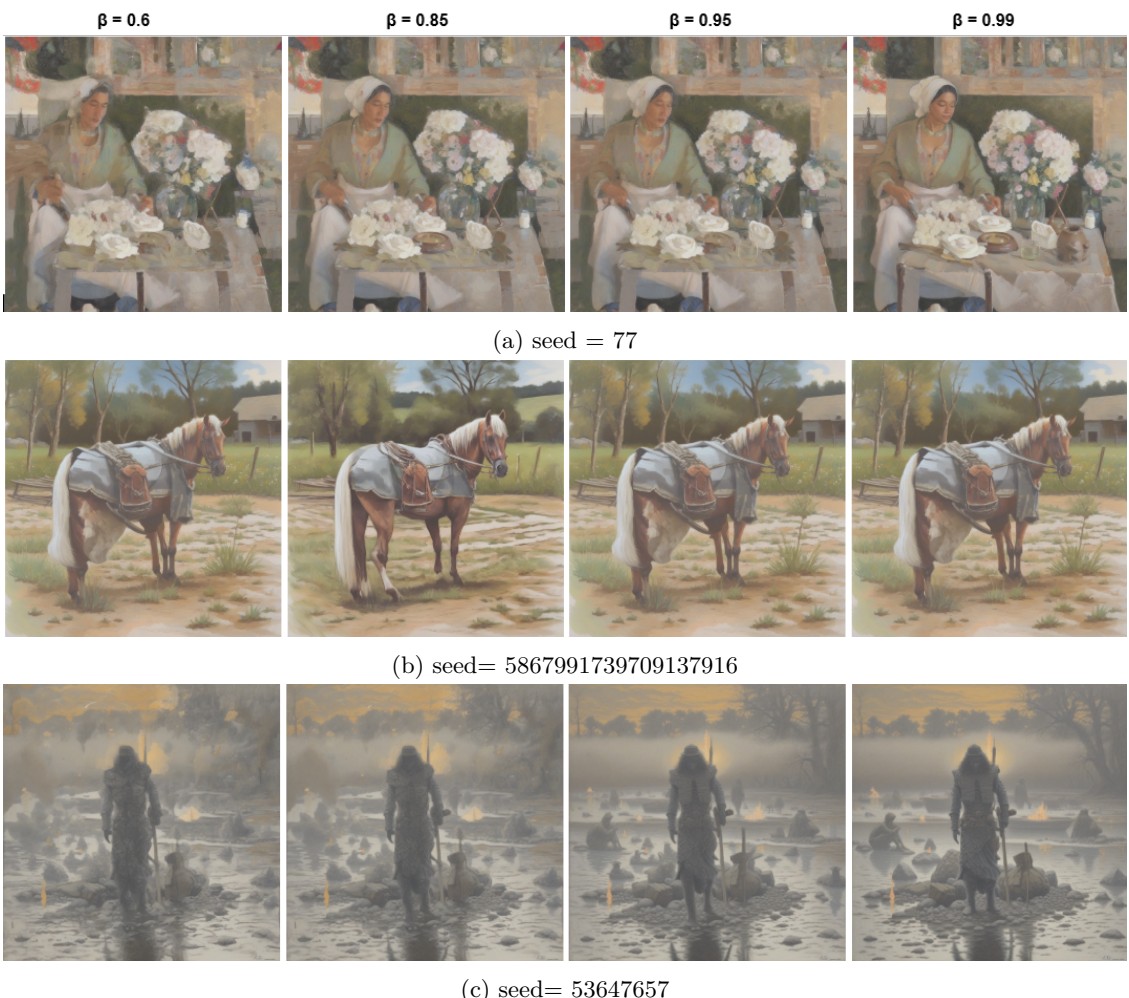

Figure 6: Unconditional images generated under SEG using EMA smoothening technique with varying $\beta$.

## 6.1 Exponential Moving Average Smoothing

The effect of varying $\beta$ on the quality of image generation is quantitatively assessed using FID scores, as presented in Table 3. Alongside these quantitative results, Figure 6 visually captures the impact of increasing $\beta$, demonstrating an improvement in image quality. These findings support the theoretical observation that higher values of $\beta$ are preferable for better alignment with the SEG framework. Furthermore, the computational efficiency of EMA-based smoothening, particularly in comparison to alternative smoothening techniques, is clearly illustrated in Figure 5 and detailed in Table 2.

## 6.2 Interpolated Box Blur

The Box Blur technique has two key parameters: $\alpha$ and kernel size. Their optimal values were determined through multiple preliminary experiments. Figure 7 presents the variation in image quality with a fixed $\alpha$ value while varying the kernel size. It is observed that at lower kernel sizes, image quality deteriorates, whereas increasing the kernel size enhances the image clarity. Similarly, Figure 8 illustrates the effect of image quality when using a fixed kernel size and varying $\alpha$. The results indicate that the image quality progressively improves as $\alpha$ increases.

The effect of varying $\alpha$ on the quality of image generation is quantitatively assessed using FID scores, as presented in Table 3. These results support the theoretical observation that higher values of $\alpha$ are preferable

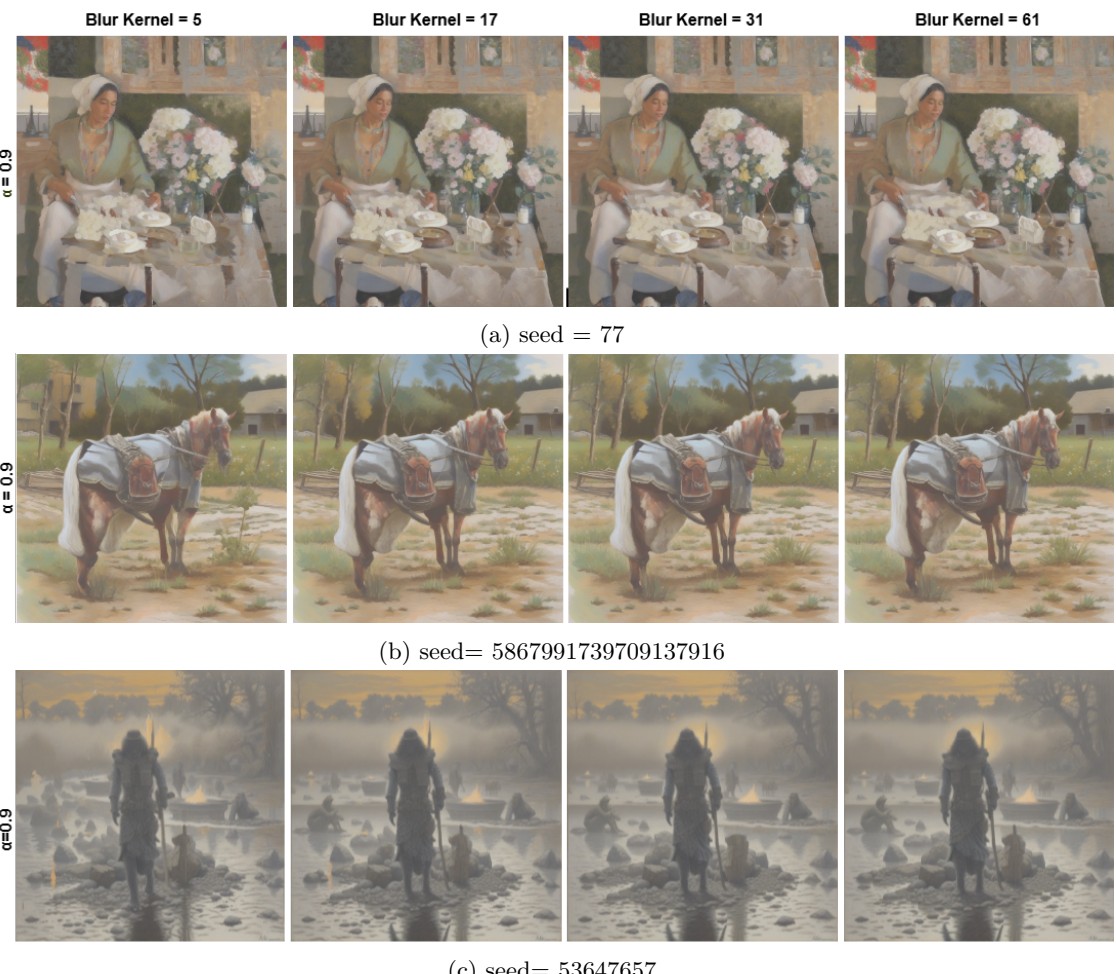

(a) seed = 77

(b) seed= 58679917397091137916

(c) seed= 53647657

Figure 7: Unconditional images generated under SEG using Interpolated Box Blur smoothing technique with varying kernel sizes (keeping $\alpha$=0.9 fixed).

for better alignment with the SEG framework. Furthermore, the computational efficiency of Interpolated Box Blur-based smoothening, in comparison to Gaussian based smoothening techniques, is clearly illustrated in Figure 5 and Table 2.

### 6.3 Why Do Our Results Deviate from the Reported Scores?

As evident Table 3, the FID scores we obtained differ significantly from those reported by the authors of the SEG paper. We attribute this deviation primarily to the following two factors:

- **Number of Inference Steps:** In our experiments, we fixed the number of inference steps to 30 across all configurations for FID evaluation. Increasing the number of steps can help reduce noise during generation, potentially leading to improved FID scores. However, the SEG authors did not explicitly mention the number of inference steps used in their evaluations, making a direct comparison difficult.

- **Choice of Attention Layers for SEG Application:** A typical UNet architecture consists of three major blocks—*down*, *mid*, and *up*—each containing multiple attention layers. In our implementation, we applied SEG smoothening exclusively to all attention layers in the *mid* section. The original paper, however, does not specify which attention layers were subjected to SEG smoothening.

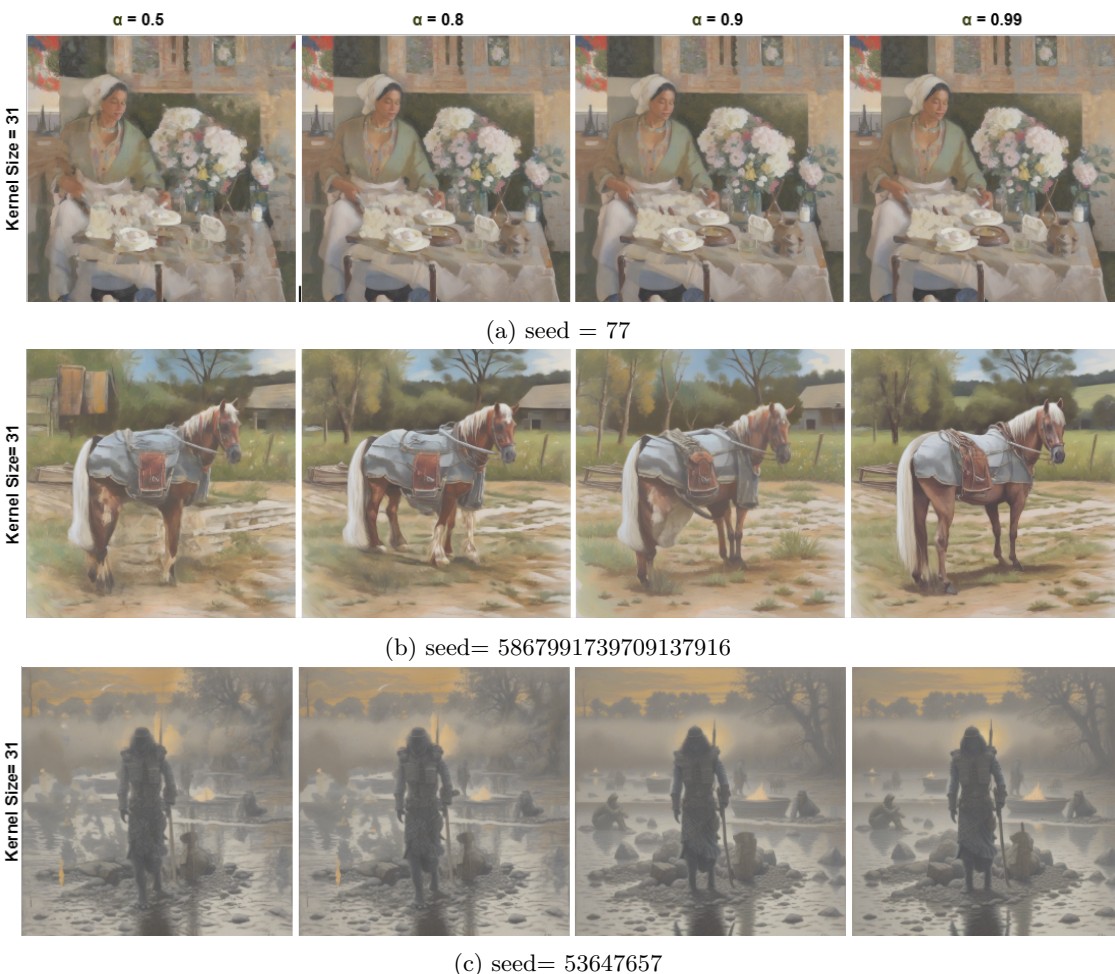

(a) seed = 77

(b) seed= 5867991739709137916

(c) seed= 53647657

Figure 8: Unconditional images generated under SEG using Interpolated Box Blur smoothing technique with varying $\alpha$ (keeping kernel size fixed).

- **Setting $\gamma_{cgf}$=0 for SEG-only Guidance** We evaluated all results with CFG completely disabled. While CFG is known to be effective with prompts, especially through the use of negative prompts, enabling it could potentially have resulted in better FID scores with the same number of inference steps.

Given these differences, further tuning—such as adjusting the number of inference steps or selectively applying smoothening to different attention layers—may help align our FID scores more closely with those reported in the original paper, or at least improve upon our current results. Nonetheless, the strong agreement observed in the FID score for the **None** case lends confidence to the validity and correctness of our evaluation pipeline.

## 6.4 Impact of Query Blurring on Attention Mechanism: Analysis Using Gradient Entropy, Laplacian Variance, and Frobenius Norm

We analyze the hidden states, represented as tensors of shape $(C, H, W)$, generated by the attention layers located in the mid-section of the U-Net architecture within the diffusion model. To quantify the effects of query blurring, we compute three distinct metrics for each channel of these hidden state activation maps and subsequently average the results across channels. The metrics employed are gradient entropy, Laplacian variance, and the Frobenius norm.

| Blurring Technique | Parameters | Reported | Ours |
|---|---|---|---|
| None | - | 129.496 | 129.4418 |
| EMA | $\beta = 0.6$ | - | 106.4188 |
| EMA | $\beta = 0.85$ | - | 105.3184 |
| EMA | $\beta = 0.95$ | - | 105.4332 |
| EMA | $\beta = 0.99$ | - | 105.6703 |
| Gaussian | kernel=3, $\sigma = 10$ | - | 116.1634 |
| Gaussian | kernel=17, $\sigma = 10$ | - | 110.3828 |
| Gaussian | kernel=31, $\sigma = 10$ | - | 109.0687 |
| Gaussian | kernel=$-1$, $\sigma = 10$ | 95.316 | 107.8214 |
| Gaussian | $\sigma \to \infty$ | 88.215 | 107.6913 |
| BoxBlur | kernel=5, $\alpha = 0.9$ | - | 114.0937 |
| BoxBlur | kernel=17, $\alpha = 0.9$ | - | 108.2538 |
| BoxBlur | kernel=31, $\alpha = 0.9$ | - | 107.1481 |
| BoxBlur | kernel=61, $\alpha = 0.9$ | - | 107.1933 |

Table 3: FID scores obtained for unconditional generation under different blurring techniques. For Gaussian blurring, a kernel size of $-1$ indicates that the kernel size was computed using Eq. 3.

The evolution of these metrics under various query blur settings across the reverse diffusion steps is depicted in Figure 9. Consistent with expectations for a denoising process, both gradient entropy and Laplacian variance exhibit a generally decreasing trend, signifying progressive feature stabilization and noise reduction. In contrast, the Frobenius norm follows a different trajectory, initially decreasing around the midpoint of the reverse diffusion process before increasing again towards the final steps. This latter behavior may be attributable to the development and refinement of intricate details within the generated image.

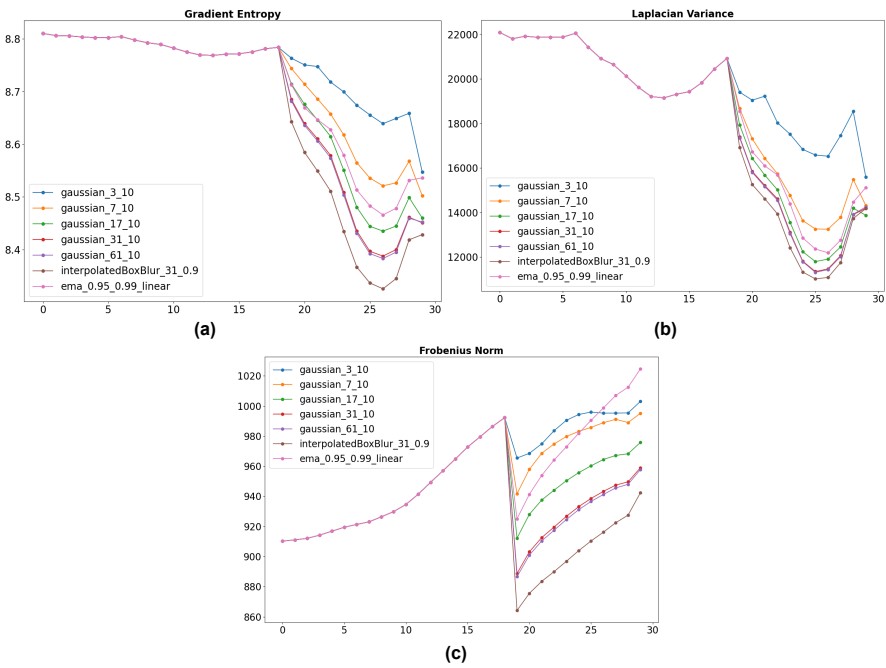

s

Figure 9: Variation of metrics like Gradient Entropy, Laplacian Variance and Frobenius Norm calculated on hidden states generated by attention layer of mid-block post attention mechanism and normalisation.

From this analysis, two primary observations warrant particular attention:

- Firstly, the trajectories defined by all three metrics for different blur settings remain remarkably coincident—identical to approximately 13 decimal places—throughout the initial stages of the reverse diffusion. Although the SEG mechanism, which influences these dynamics, is applied only up to step 10 of the reverse process (refer to Table 1), significant divergence between the trajectories corresponding to different blur settings does not manifest until approximately step 17. This notable delay suggests a subtle, perhaps cumulative, downstream effect resulting from the initial query blurring applied much earlier. This phenomenon is discussed in detail in Appendix A.2.

- Secondly, these quantitative metrics serve as effective indicators of image generation quality. The trends observed align consistently with established measures like FID scores (Table 3) and with qualitative assessments via visual inspection. Figure 9 offering comparison among various smoothening techniques, out of which Interpolated Box Blur stands out.

## 7    Conclusion

This study validates the findings of the SEG paper, demonstrating that energy smoothing through attention weight blurring enhances image generation quality while decoupling it from external guidance like CFG. Our analysis shows that subtle energy guidance minimally alters the generation trajectory, ensuring realistic outputs without artifacts. We further explored more efficient smoothing techniques, highlighting the importance of precise early-stage guidance. This approach not only improves generation quality but also suggests future research into its role in accelerating convergence during reverse diffusion, potentially reducing the required denoising steps. While pursuing these technical advancements, we recognize the importance of considering the broader context of generative AI. Like other generative models, the techniques discussed could be subject to societal concerns regarding potential misuse, inherent biases, fairness, and environmental impact. Addressing these ethical considerations responsibly is crucial for the continued development of the field.

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

# A  Appendix

## A.1  Justifying Alternative Smoothing for SEG

### A.1.1  Theoretical Alignment of Interpolated Box Blur with SEG Principles

As established in Section 5.1, the effectiveness of a blur operator within the Smoothed Energy Guidance (SEG) framework Hong et al. (2024) relies on satisfying key mathematical properties related to smoothing the attention energy landscape. Here, we demonstrate that the Interpolated Box Blur method used in this work (defined in Section 5.1.3 by Eq. 10 with $\alpha$ close to 1) fulfills these requirements:

1. **Mean Preservation:** The input tensor $I$ clearly has mean $\mathbb{E}[I]$. Standard box blur $I'$ also preserves this mean, $\mathbb{E}[I'] = \mathbb{E}[I]$. The output $Img' = (1 - \alpha)I + \alpha I'$ is a linear combination with weights summing to 1. Therefore, the mean is preserved:

$$\mathbb{E}[Img'] = (1 - \alpha)\mathbb{E}[I] + \alpha\mathbb{E}[I'] = (1 - \alpha)\mathbb{E}[I] + \alpha\mathbb{E}[I] = \mathbb{E}[I]$$

2. **Variance Reduction:** Standard box blur $I'$ acts as a low-pass filter and significantly reduces variance compared to the original input $I$, $Var[I'] < Var[I]$. The variance of the interpolated output $Img'$ is given by $Var[(1-\alpha)I+\alpha I']$. Since $\alpha$ is close to 1 (e.g., 0.9), the output is strongly influenced by the low-variance $I'$ component. While the exact variance depends on the covariance between $I$ and $I'$, the strong weighting towards $I'$ ensures a substantial reduction in variance compared to the original $I$, satisfying the core requirement $Var[Img'] \leq Var[I]$.

3. **Kernel Weight Constraint ($b_{ii} < 1$):** The Interpolated Box Blur operation can be represented by an effective linear operator $B_{interp} = (1 - \alpha)I_{op} + \alpha B_{box}$, where $I_{op}$ is the identity operator and $B_{box}$ represents the box blur convolution matrix. The diagonal elements $b_{interp,ii}$ of this effective operator correspond to the weight of an input element $I(x, y)$ contributing to the output $Img'(x, y)$ at the same location. This is calculated as:

$$b_{interp,ii} = (1 - \alpha) \cdot (\text{diag element of } I_{op}) + \alpha \cdot (\text{diag element of } B_{box})$$

$$b_{interp,ii} = (1 - \alpha) \cdot 1 + \alpha \cdot (1/k^2) = (1 - \alpha) + \alpha/k^2$$

where $k$ is the box kernel size. Given that $\alpha$ is close to 1 (e.g., $\alpha = 0.9$) and typical kernel sizes are $k \geq 3$, the value is significantly less than 1. For $\alpha = 0.9, k = 3$, $b_{interp,ii} = 0.1 + 0.9/9 = 0.2$. Since $0 \leq b_{interp,ii} < 1$, this crucial condition derived from the Hessian analysis in (Hong et al., 2024, Appendix A.3) is met.

By satisfying these three essential properties (mean preservation, variance reduction, and $b_{ii} < 1$), Interpolated Box Blur aligns with the theoretical requirements for smoothing operators within the SEG framework. This demonstrates its capability to effectively attenuate the attention energy landscape curvature, justifying its use as an alternative to Gaussian blur.

### A.1.2  Theoretical Alignment of EMA Smoothing with SEG Principles

We analyze the specific Exponential Moving Average (EMA) implementation used in this study (see Section 5.1.2), applied along the token dimension ('dim=2') of the query tensor, against the generalized requirements for Smoothed Energy Guidance (SEG) operators outlined in Section 5.1 and derived from Hong et al. (2024). The implementation corresponds to the recursive definition $\text{EMA}_t = \beta x_t + (1 - \beta)\text{EMA}_{t-1}$ with $\beta$ close to 1 (e.g., 0.85 or 0.95).

1. **Mean Preservation:** A linear filter preserves the mean if its weights sum to 1. The effective weights in the vectorized implementation correspond to $w_j = \beta(1 - \beta)^j$ for $j = 0, ..., T - 1$. The sum of these weights over a finite sequence of length $T$ is:

$$S = \sum_{j=0}^{T-1} \beta(1 - \beta)^j$$

This is the sum of a finite geometric series with first term $\beta$, ratio $(1 - \beta)$, and $T$ terms:

$$S = \beta \frac{1 - (1 - \beta)^T}{1 - (1 - \beta)} = \beta \frac{1 - (1 - \beta)^T}{\beta} = 1 - (1 - \beta)^T$$

Since $0 < \beta < 1$ and $T \geq 1$, the term $(1 - \beta)^T$ is positive and less than 1. Therefore, the sum $S = 1 - (1 - \beta)^T$ is strictly less than 1. Thus, the EMA implementation **does not strictly preserve the mean**. However, as our implementation uses $\beta$ close to 1, the term $(1 - \beta)$ is close to 0. Consequently, $(1 - \beta)^T$ becomes negligible very quickly for typical sequence lengths $T$, making $S \approx 1$. The mean is therefore preserved to a decent degree of approximation in practice.

2. **Variance Reduction:** EMA acts as a low-pass filter by averaging values with exponentially decaying weights $((1 - \beta)^j)$. Applying it along the token dimension smooths variations across the sequence. Therefore, this EMA implementation **satisfies the variance reduction criterion**, $Var[\text{EMA}(I)] \leq Var[I]$.

3. **Kernel Weight Constraint ($b_{ii} < 1$):** This constraint requires that the direct contribution of an input element $I_t$ to its corresponding output element $\text{EMA}_t$ is weighted by less than 1. In the effective convolution implemented, the weight applied to the input element $q_t$ when computing the output element $\tilde{q}_t$ is $\beta(1 - \beta)^t$ (based on analysis of the specific 'cumsum' and 'flip' operations, resulting in an anti-causal filter structure as implemented). Since $0 < \beta < 1$, this weight is always less than 1 for $T \geq 1$. Considering the recursive form, the weight on the current element $x_t$ is $\beta$. Since $\beta < 1$, the implementation **aligns with the $b_{ii} < 1$ requirement**.

In summary, the vectorized EMA implementation used satisfies the variance reduction and kernel weight constraints required for SEG smoothing operators. While it does not strictly preserve the mean due to the weights summing to $1 - (1 - \beta)^T$, the deviation from 1 is negligible for the chosen values of $\beta$ close to 1 (corresponding to $\alpha \approx \beta$ close to 1 in the code parameterization). This suggests that EMA provides the necessary smoothing effect to attenuate the energy landscape curvature, making it empirically viable for SEG, although it represents a slight theoretical departure from operators that strictly preserve the mean.

### A.2 Detailed Mathematical Analysis of the Amplification Mechanism

In this appendix, we provide a detailed derivation of the exponential amplification of minute perturbations in the reverse diffusion process. This phenomenon—where global metrics initially fail to capture any differences (showing identical values up to 13 decimal places, see Figure 9) and then diverge—illustrates a butterfly effect analogous to that observed in chaotic systems.

### A.3 Reverse Diffusion Update Function

We define the reverse diffusion update at timestep $t$ as

$$x_{t-1} = f_t(x_t) = \frac{1}{\sqrt{\alpha_t}} \left( x_t - \frac{1 - \alpha_t}{\sqrt{1 - \bar{\alpha}_t}} \epsilon_\theta(x_t, t) \right), \tag{16}$$

where $\alpha_t$ is determined by the noise schedule and

$$\bar{\alpha}_t = \prod_{i=1}^{t} \alpha_i,$$

and $\epsilon_\theta(x_t, t)$ denotes the noise prediction by the neural network.

### A.4 Linearization via Taylor Expansion

Let $x_t^{(1)}$ and $x_t^{(2)}$ be two hidden states at time $t$ corresponding to different blurring settings. Their difference is given by

$$\delta_t = x_t^{(1)} - x_t^{(2)}, \tag{17}$$

with $\|\delta_t\|$ on the order of $10^{-13}$. For a sufficiently small $\delta_t$, we can approximate the effect of $f_t$ by a first-order Taylor expansion around $x_t^{(2)}$:

$$f_t(x_t^{(2)} + \delta_t) \approx f_t(x_t^{(2)}) + J_t\,\delta_t, \tag{18}$$

where the Jacobian $J_t$ is defined as

$$J_t = \frac{\partial f_t}{\partial x_t}\Big|_{x_t = x_t^{(2)}}. \tag{19}$$

Thus, the difference at timestep $t-1$ becomes

$$\delta_{t-1} = f_t(x_t^{(1)}) - f_t(x_t^{(2)}) \approx J_t\,\delta_t. \tag{20}$$

### A.5  Iterative Amplification of Perturbations

By applying the linearized update repeatedly over $k$ iterations, the difference evolves as

$$\delta_{t-k} \approx \left( \prod_{i=t-k+1}^{t} J_i \right) \delta_t. \tag{21}$$

Taking the norm on both sides, we obtain

$$\|\delta_{t-k}\| \approx \left\| \prod_{i=t-k+1}^{t} J_i \right\| \|\delta_t\|. \tag{22}$$

In systems exhibiting sensitive dependence on initial conditions, the product of the Jacobian norms grows exponentially. That is, we can approximate

$$\left\| \prod_{i=t-k+1}^{t} J_i \right\| \approx e^{\lambda k}, \tag{23}$$

with $\lambda > 0$ acting as an effective Lyapunov exponent. Thus, we arrive at the central amplification equation:

$$\|\delta_{t-k}\| \approx e^{\lambda k}\|\delta_t\|. \tag{24}$$

### A.6  Interpretation and Relation to Global Metrics

The derivation shows that, although the global metrics (Frobenius norm, Laplacian variance, and entropy) initially fail to capture any differences in the hidden states—reporting nearly identical values (up to 13 decimal places) across all blurring settings—the nonlinear dynamics of the reverse diffusion process amplify even infinitesimal differences. This exponential growth, captured by the effective **positive Lyapunov exponent ($\lambda > 0$)**, explains the eventual divergence of the metric curves observed in later iterations. In essence, the system exhibits a butterfly effect, where the small, undetectable perturbations present in the early iterations become significantly amplified over time, leading to distinct final outcomes.

## B  Generalization Across Different Random Seeds

To further validate the robustness of our findings, we present additional qualitative results generated using the same experimental settings but with few more random seeds.

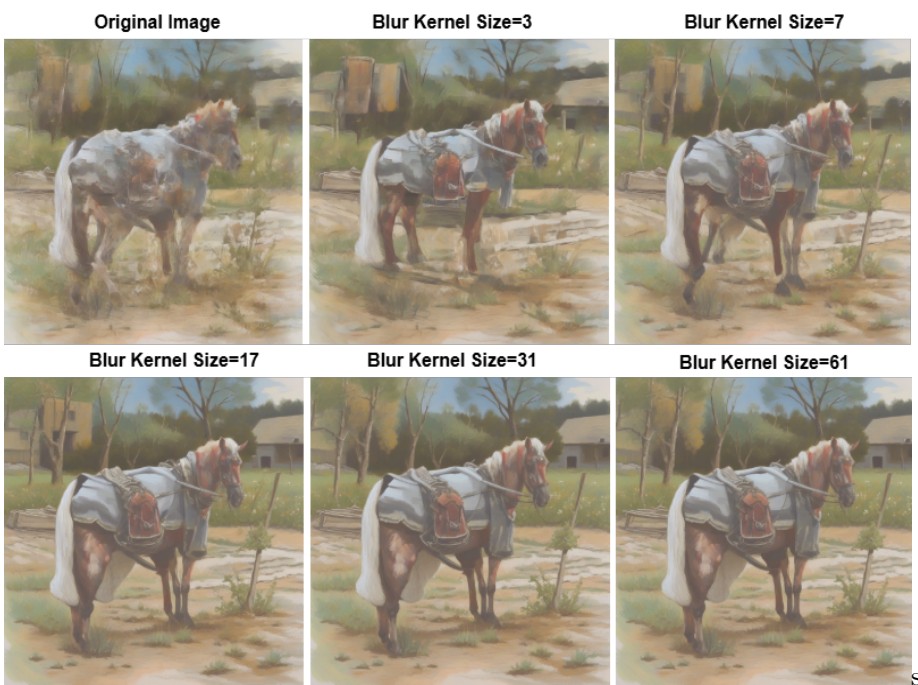

Figure 10: Image quality assessment using Gaussian blur with varying kernel sizes, where $\sigma$=10. Blurring was applied to the mid-attention layers and during the initial 1/3 of the total iterations with reproducibility Seed(5867991739709137916)

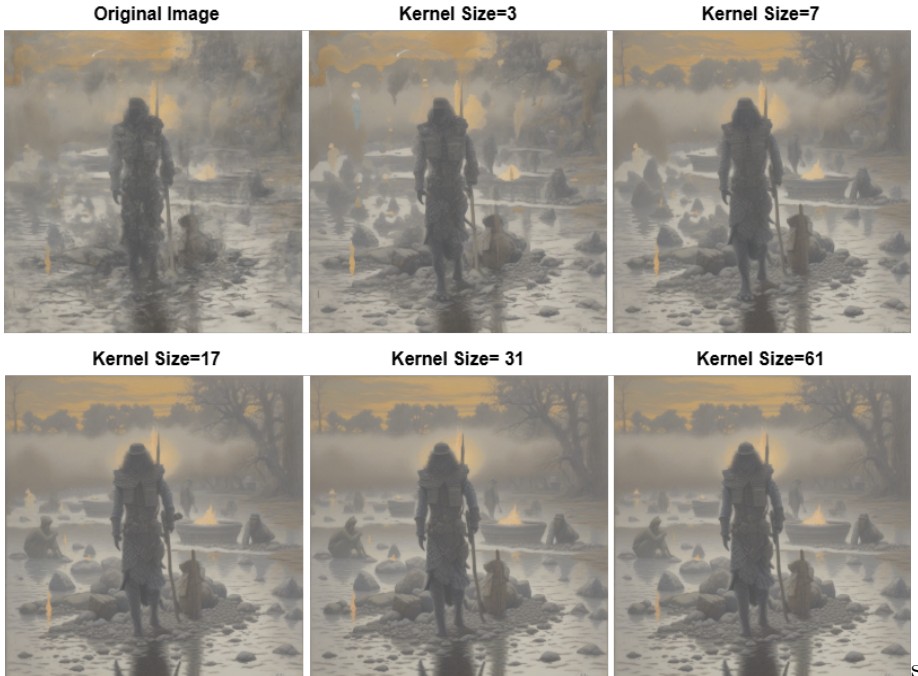

Figure 11: Image quality assessment using Gaussian blur with varying kernel sizes, where $\sigma$=10. Blurring was applied to the mid-attention layers and during the initial 1/3 of the total iterations with reproducibility Seed(53647657)

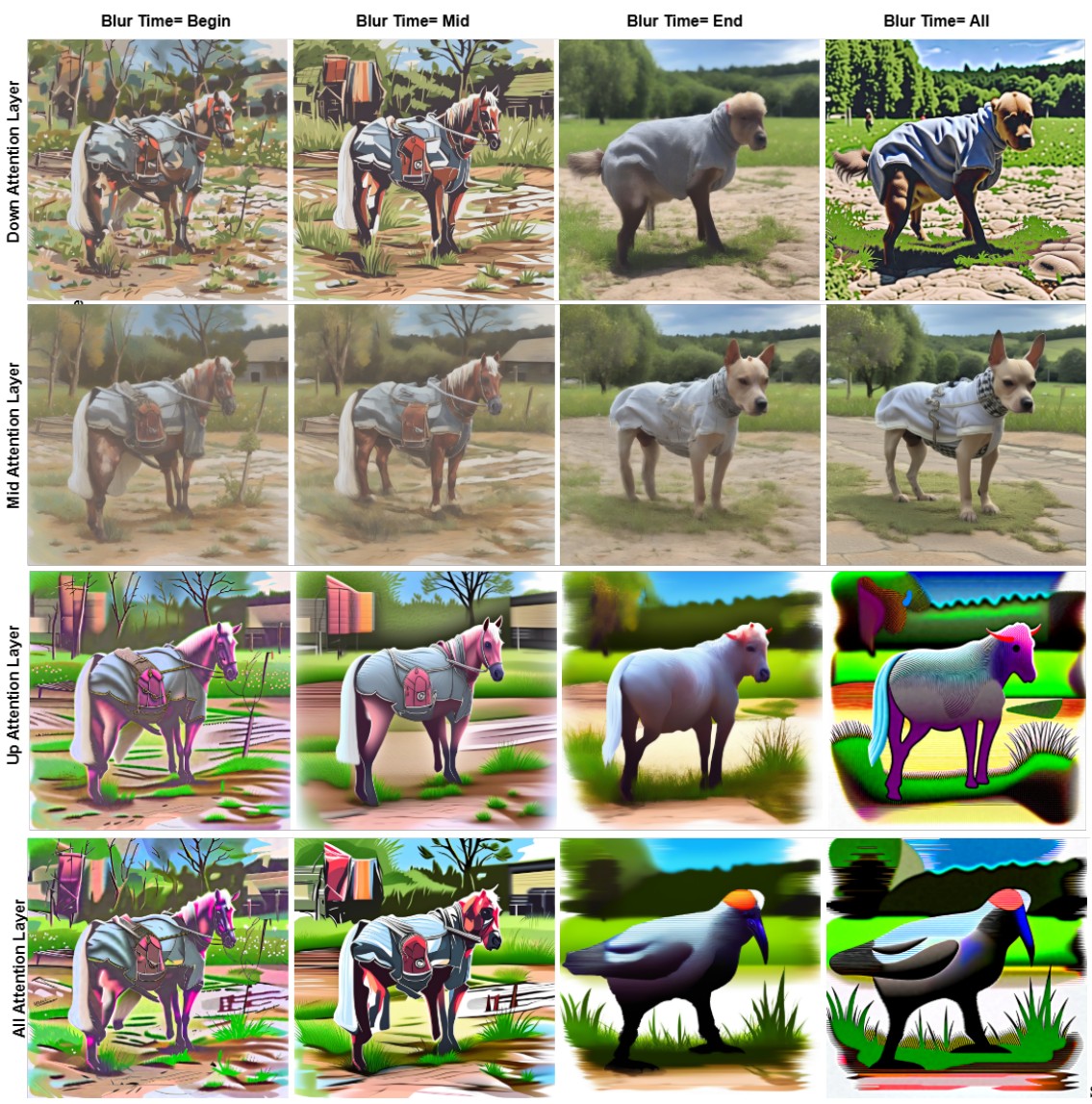

Figure 12: Image quality generation for a fixed kernel size of 31 and $\sigma = 10$. The 30 iterations of reverse diffusion are divided into three equal phases: Begin (iterations 1-10), Mid (iterations 11-20), and End (iterations 21-30), ensuring reproducibility with Seed (58679917397091379116).

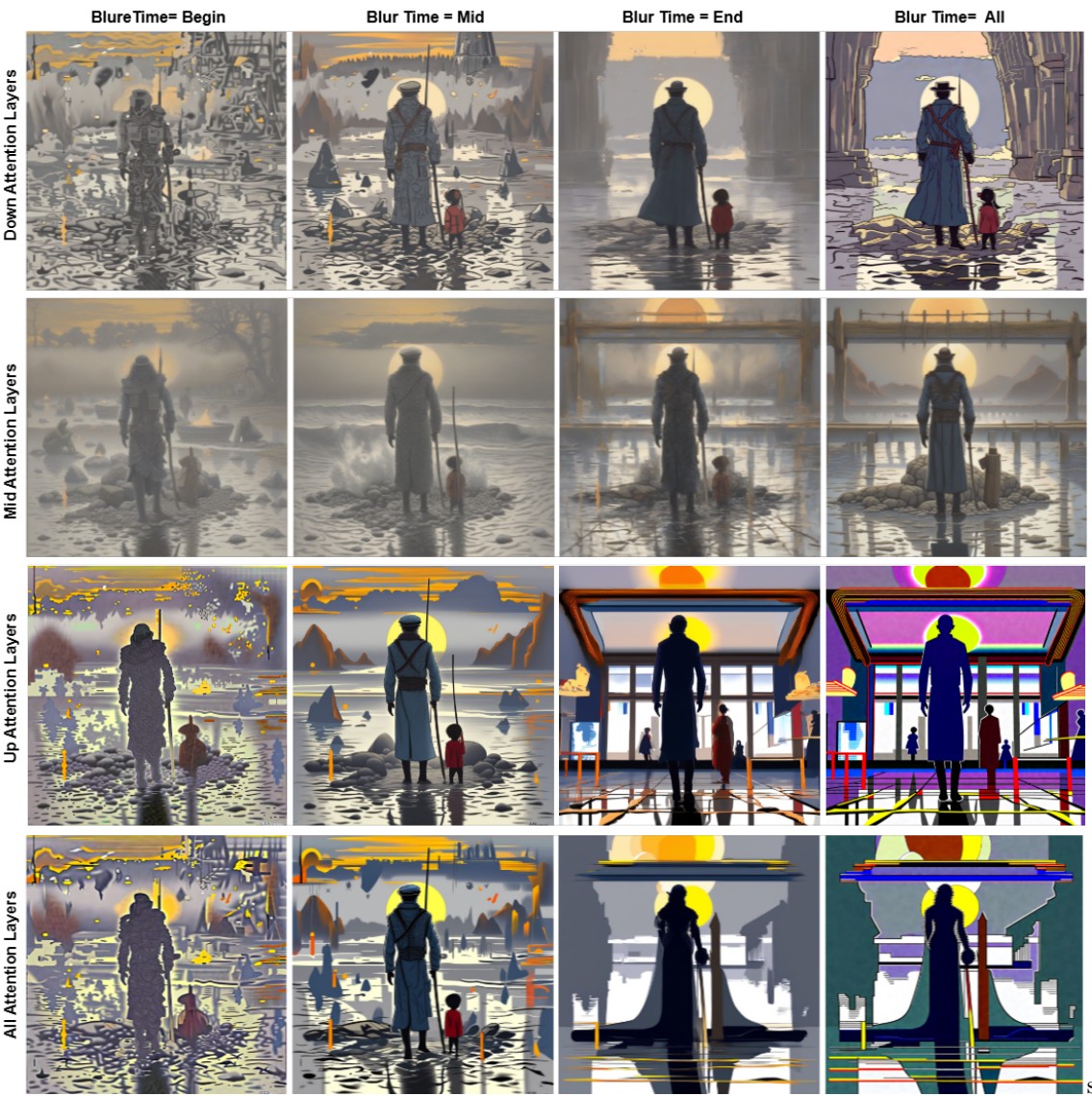

Figure 13: Image quality generation for a fixed kernel size of 31 and $\sigma = 10$. The 30 iterations of reverse diffusion are divided into three equal phases: Begin (iterations 1-10), Mid (iterations 11-20), and End (iterations 21-30), ensuring reproducibility with Seed (53647657).

