# OpenReview forum: "[Re] Smoothed Energy Guidance for Diffusion Models"
_TMLR — Rejected by TMLR_

### Review · Reviewer_rdvn · 2025-03-17

**Summary Of Contributions:**

This manuscript extends the work of Hong (NeurIPS 2024) on “Smoothed Energy Guidance: Guiding Diffusion Models with Reduced Energy Curvature of Attention” (SEG).

Background: The original NeurIPS 2024 work (hereafter SEG) introduces a diffusion model guidance mechanism similar to classifier-free guidance (CFG), but uses a prediction with blurred attention maps as an “anchor” prediction for guidance, rather than an unconditional prediction. The SEG work discusses the flexibility (can decide blurring standard deviation, usable in combination of CFG, compatible with other works such as ControlNet) and advantages over CFG (usable for unconditional generation, less artifacts) of their method. The original work also provides a theoretical foundation of their method, linking the attention mechanism to a minimization step of some energy function, and linking SEG to a smoothed version of this energy function.

The reviewed manuscript builds upon SEG by investigating the following aspects:

C1/ [Kernel size for Gaussian blurring] The authors claim that the kernel size of 61x61  in the original SEG work to generate the Gaussian blur filter (with $\sigma=10$ as suggested in SEG paper) is “computationally impractical”. They claim that a kernel size of 31x31 “yield almost similar image quality”, avoiding the “significantly escalating computational costs” of going from  31x31 to 61x61.

C2/ [Layer selection] The authors claim that the selection of layers of the U-Net at which to apply SEG hasn’t been reported in the original paper. They claim that applying SEG to layers other than the bottleneck/middle layers “severely degrades image quality.”

C3/ [Disable SEG at low timesteps] The authors claim that applying SEG to every inference step “significantly increases computational costs”. They propose restricting SEG to only the first third of inference steps, claiming it is sufficient.

These 3 contributions (C1, C2, C3) are supported only qualitatively through a single image generation example (one fixed seed, one single blurring standard deviation value, single guidance scale, without prompt, without ControlNet). One additional qualitative example is included in the appendix. However, no quantitative metrics (image quality) or theoretical justification are provided. The contributions on significant “computational costs”/“computationally impractical” are not evidenced as inference time is never measured.

Then, the manuscript explores two further aspects:

C4/ [Alternative blurring techniques] The authors also briefly discuss 2 alternative blurring techniques, Box Blurring and EMA Blurring, claiming they have “more efficient” “computational complexity”. However, the description of EMA Blurring is incomplete, and the corresponding code seems commented out (“def ema_smoothing_time_dependent”). The claim that Box Blurring is computationally more efficient than Gaussian Blurring lacks justification, particularly given the dependence on kernel size.

C5/ [Butterfly effect] The authors also discuss some “butterfly effect” present in diffusion models. I find its relevance to SEG is unclear, the chosen metrics and their implications are not well-justified, and I don't understand the significance of these observations.

While the manuscript has interesting findings and computational considerations for SEG, the manuscript currently lacks quantitative validation, theoretical analysis, broader experimental evaluation, discussions of related works, etc.

**Audience:**

Yes

**Broader Impact Concerns:**

The reviewed work does not include a Broader Impact Statement section. As with any work on generative models, there are concerns / societal impacts / ethical considerations (potential misuses, bias, fairness, environmental impact of image generation), although they are not specific to this work. A short acknowledgement of these concerns should be sufficient in my opinion.

**Claims And Evidence:**

No

**Requested Changes:**

See the weaknesses above, this manuscript appears to be a work-in-progress rather than a fully developed submission, particularly considering the standards of TMLR.

The most critical requested changes are as follows:

R1/ Provide empirical evidence (measuring inference time, memory usage, CLIP score, image quality, over many generations, with standard errors ; qualitative results on many seeds) for the claims, a single image generation is not enough. See W1 above

R2/ Clarify the contributions and insights. See W2, W3, W5, W6, W7 above

R3/ The paper does not relate sufficiently with prior research. The authors should better discuss the original SEG work and the prior works that relate to their additional experiments. See W4 above.

Addressing these changes would strengthen the paper without securing my recommendation for acceptance.

**Strengths And Weaknesses:**

Strengths

S1/ [Relevant work] The experiments and findings made by the authors are globally of interest, especially regarding alternative blurring techniques for SEG, reducing computational cost of SEG, and studies of which layers to apply SEG, although they are a bit niche (SEG is not widely used). These offer a few insights for more efficient implementations.

S2/ [Additional insights on top of SEG] The studies in this manuscript indeed provide experiments and insights on top of the original paper. In fact, the current manuscript does not reproduce any result from the original paper, but only additional ablation studies / alternative studies.


Weaknesses

W0/ Provided code (anonymous Github) does not work. Running “. scripts/run.sh” returns the following error:
SEGCFGSelfAttn.py, line 19
ImportError: cannot import name 'ema_smoothing_time_dependent' from 'diffusion_utils.blur'

The corresponding code for the function 'ema_smoothing_time_dependent' is commented out. Fixing this issue gives another error:
SEGCFGSelfAttn.py, line 19
ImportError: cannot import name 'alpha_increasing' from 'diffusion_utils.blur'


W1/ [Claims are not sufficiently validated] There is a clear lack of empirical validation in this manuscript.
- These 3 contributions (C1, C2, C3) are supported only qualitatively through a single image generation example (one fixed seed, one single blurring standard deviation value, single guidance scale, without prompt, without ControlNet). One additional qualitative example is included in the appendix.
- No quantitative evaluation is provided (e.g., FID, CLIP similarity, standard error across multiple seeds). Claims about computational efficiency (e.g., “computationally impractical”) are not substantiated with actual runtime or memory measurements.
- No trade-off, e.g., balancing computational savings and image quality degradation, is analyzed.
- It is not clear whether the theoretical foundation of SEG still applies, for the alternative blurring techniques, or reduced kernel size (truncated gaussian blur filter?). Does it still lead to an energy function with reduced curvature, or is it necessary to use gaussian blurring for this?
- The findings, e.g. “almost similar image quality” are substantiated by only 1 qualitative evaluation (1 generation), and the claimed observations are not always clear (e.g., why kernel size of 31 and not 17?)


W2/ [Paper is not self sufficient and writing/notation/background need be improved]
The paper is not self-sufficient. The description of the original SEG method is incomplete. The paper does not say which diffusion model is used (no mention of “Stable Diffusion” in the manuscript). Notations need to be better introduced. It is necessary to mention we are considering a U-Net with attention layers before introducing equation 1. It would also help to say here that the blurring is only applied to the mid-block layers of the U-Net in the SEG work. It is necessary to introduce classifier free guidance and energy based guidance. The paper cited for energy based guidance at the bottom of page 1 seems unrelated. Stating the dimensions of Q, K, V in Eq. (1) would also help. The connection to the energy function is not well explained.


W3/ [Contributions could be better articulated] The contributions and insights are not clearly articulated. In Section 1.2, the paper states that it focuses on C1 (blur kernel size), C2 (layer selection), and C3 (inference step frequency). However, it later introduces C4 (alternative blurring techniques) and C5 (butterfly effect) without properly integrating them with C1/C2/C3. Also, the introduction (Section 1.2) states the focus is to study these points, but the main findings and insights are not explicitly summarized as I wrote in C1, C2, C3 above, making it difficult to extract actionable findings in the first readings.


W4/ [No discussion of related works for contextualization]
- The manuscript does not cite the original SEG paper.
- The paper includes only 8 references, missing key literature on:
* CFG schedulers: Similar to how the authors propose to disable SEG after some steps (“2.3 Minimizing Redundant Computations in Blurring Iteratively”), the same method of disabling or reducing CFG after some steps was proposed and discussed in prior works [1, 2, 3, 4, 5, 6].
* Perturbation-based guidance: As the blurring technique changes, it is unclear if the theoretical foundation in the SEG work still holds. Random perturbations to guide the model were mentioned in previous literature [7, 8, 9], and it would be necessary to discuss and relate to these works.
* As one of the goals is to reduce computational cost, many related works that improve computational cost of diffusion models (such as consistency distillation, guidance distillation, adversarial diffusion distillation, etc) should also be discussed to nuance the discussion (saving one third of the NFEs before distillation might not be relevant nowadays, as the models are later distilled).


W5/ [Writing and clarity issues]
Several claims lack precision, for example:
- kernel size 61x61 is “computationally impractical” -> No evidence. How much time does it use to blur an image with a kernel size 61 vs 31? Also, I believe SD uses low-resolution thanks to the VAE processing (64x64?), and attention maps in middle layers are of even lower resolution (8x8?). Can you discuss the relevance of kernel sizes above 8x8?
- “There is ambiguity regarding the specific set of attention layers” -> It is not ambiguous in the original paper, where they say “To sample with SEG, we choose the same attention layers (mid-blocks) and guidance scale as PAG [1].”
- “we introduced flexible hyperparameter settings, allowing adjustments to σ” -> It is mentioned earlier the original method also allows adjustments to σ.
- “during the initial 30-35% of total iterations” -> Why is it a range between 30 and 35%? Why not say exactly the number of steps?
- It is very unclear why in the setup “blur time all” gives the exact same generated image regardless at which layer the blurring is applied (Figure 2) -> It is hard to trust these results.
- The discussion on "a mathematically equivalent perspective on the linear combination of different score models" (Page 5) merely rewrites an existing formula (going from Eq. 5 to 6) without adding meaningful interpretation or justification. The analogy to momentum-based optimizers like Adam in Figure 3 is not substantiated at all: the figure simply draws 6 arrows without any clear link to momentum-based updates. The claim that "energy guidance functions similarly to momentum" needs explicit mathematical reasoning or experimental validation. I don’t understand how the drawing with 6 arrows supports this. I also do not understand how this section relates to the rest of the work.
- The equations for EMA are not clear in section 2.4.1. -> What is $x_t$ and $EMA_t$? Where is it used and what for?
- Box Blur in section 2.4.2 -> Why interpolate the blur and original with this method and not with the original one?

W6/ Presentations issues
- Figures are not well-integrated with the text, making it difficult to follow experiments.
- Typo(s), e.g., “Blure” instead of “Blur”

W7/ [Misleading title, lack of reproduction of original experiments] The manuscript title is exactly the same as the original work SEG. This is very confusing. Also the manuscript does not attempt to reproduce or quantitatively validate the experiments from the original SEG paper. The term “reproducibility” and framing of the work (eg conclusion: “This study validates the findings of the SEG pape”) is misleading, as the manuscript does not focus on verifying or reproducing the results of the original SEG paper but instead presents modifications and optimizations.

W8/ [Potential anonymization issues] “Anonymous Github” code (hyperlink in the abstract) seems not anonymized properly, as potentially identifiable information remains in the files. For instance: pyproject.toml (potential email address of the authors) and metric_visualization.ipynb (path reveals the potential username of the authors)




[1] Castillo, Angela, et al. "Adaptive guidance: Training-free acceleration of conditional diffusion models." arXiv Dec 2023.

[2] Wang, Xi, et al. "Analysis of Classifier-Free Guidance Weight Schedulers." arXiv April 2024 + TMLR 2024.

[3] Kynkäänniemi, Tuomas, et al. "Applying guidance in a limited interval improves sample and distribution quality in diffusion models." arXiv April 2024 + NeurIPS 2024.

[4] Chang, Huiwen, et al. "Muse: Text-to-image generation via masked generative transformers." arXiv Jan 2023 + ICML 2023.

[5] Sadat, Seyedmorteza, et al. "CADS: Unleashing the diversity of diffusion models through condition-annealed sampling." arXiv Oct 2023 + ICLR 2024.

[6] Yan, David, et al. "Animated Stickers: Bringing Stickers to Life with Video Diffusion." arXiv Feb 2024.

[7] Karras, Tero, et al. "Guiding a diffusion model with a bad version of itself." Advances in Neural Information Processing Systems 37 (2024): 52996-53021.

[8, 9] PAG and SAG papers, as cited in the original SEG work.

---

> ### Author Response · Authors · 2025-05-04
> **FID score evaluation, Improved Presentation, Extensive Literature Review, Code Debugging**
>
> We have submitted the revised version of our manuscript, trying to incorporate all your concerns, sequentially addressing your concern in same order below:
> - To address the concerns raised as **"W0"**, we have updated the readme for easier code setup, essentially, the error you faced was because of not including current dir in PYTHON_PATH. The small bugs regarding EMA code have also been sorted out in updated Anonymous GitHub code.
> - Addressing concern **"W1"** , the claims are now reinforced with more seeds, more values for energy guidance strength and more gaussian blurring strength. To compare the techniques quantitatively, Fréchet Inception Distance (FID) for each smoothening technique with its varying params have been incorporated. The concerns regarding the computational burden are more strongly reinforced with the inclusion of runtime statistics of various techniques for energy guidance. The mathematical alignment of newer proposed smoothening techniques like EMA, BoxBlur with the SEG Framework have also been studied and included in appendix.
> - Improving on **"W2"**, we have included a table dedicated to explaining the various hyperparams involved in this study of Energy Guidance, clearly addressing the concern regarding the diffusion model used, which layers of Unet was this energy smoothening applied, trying to make our manuscript self-complete.
> - Addressing **"W3"**, we admit our mistake of improper placement of different or unrelated sections, resulting in bad impression in the reader's mind. In our updated submission, we have segregated the study of alternative blurring techniques and butterfly effect from the main claims (C1, C2, C3), and dedicated new section : **Experiments Beyond Paper**, to reduce the ambiguity.
> - **"W4"** : One of the most significant oversights in our previous submission was the omission of the literature review. Although it was prepared, it was unintentionally left out—we sincerely regret this mistake and have now included it in the updated version.
> - Addressing ambiguity regarding **"W5"**, even though UNet works in latent space and attention maps are of shape (64x64, 32x32), but each query tensor (Q) which is to be blurred has shape : (Batch , AttnHeads, Token, Embedding)... so, even though Token (1024(=32x32) or 4096(=64x64)), there are many of them, indetail discussion and differences in runtime under different blurring techniques have been done in updated submission. We have also addressed the ambiguity regarding the blurring schedule (30-35%) clearly stating the schedule we adopted (dividing the num_inference_steps in 3 parts). The similarity between guidance in diffusion model and momentum in optimizers is also reinforced through better explanation and the concept of orthogonality to main trajectory. Ambiguity regarding EMA equation and interpolation in BoxBlur is also addressed in updated manuscript.
> - Imporving on **"W6"**, A more thorough grammar and spelling check is used in the updated submission. Also, figures are integrated with their place of discussion for easier follow-up and better readability.
> - Addressing the **"W7"**, we have changed the title of our reproducibility work taking inspiration of past reproducibility papers. This is our first submission, hence could not recognize this failure. We have reproduced the FID scores for unconditional generation and compared to that reported in the SEG paper, apart from extra research which is subtly orthogonal to the direction of reproducibility.
> - For addressing the **"W8"**, we have created a new anonymized link of our GitHub code, completely removing all sources of identity leak which could violate the submission guidelines.
>
> Working on all these reviews, taught me newer insights and best practices. I sincerely apologize for minor but unexpected mistakes done from our side. But also really thankful for guiding through the depth of how research work is done.

---

### Review · Reviewer_6pCH · 2025-04-01

**Summary Of Contributions:**

This work provides an empirical investigation of Smoothed Energy Guidance (SEG) for diffusion models [1]. The authors set out to reproduce findings and provide additional ablation experiments over SEG hyperparameters, thereby, improving the performance of the method. They consider the task of image generation to conduct this ablation study and use visual assessment of images to determine reasonable hyperparameter values (ranges). The authors also test how inference iteration affects the performance of SEG and demonstrate that guidance in the early stages of generation is more beneficial. Through this ablations study, the authors propose two new approaches to help improve SEG: (1) Exponential Moving Average (EMA) and (2) BoxBlur. The authors evaluate EMA and BoxBlur through a visual assessment of image generation quality.

[1] Hong, S., 2024. Smoothed energy guidance: Guiding diffusion models with reduced energy curvature of attention. arXiv preprint arXiv:2408.00760.

**Audience:**

Yes

**Broader Impact Concerns:**

I have no broader impact concerns.

**Claims And Evidence:**

No

**Requested Changes:**

- Add content to the manuscript describing and outlining the details of the experiments, i.e. models, tasks, hyperparameters (see Weaknesses).
- More thorough empirical experiments.
    - For experiments in Figure 7, it would be useful to see more seeds and error bars.
    - It appears for the qualitative results, only 2 seeds are considered. Showing results across more seeds (and in the main body of the text) is pertinent to providing sufficient evidence for the claims of this work.
    - Addition of quantitative evaluation(s) of image results/quality. Without quantitative evaluation, it is difficult to assess whether the claims of this work are sufficiently supported. Depending on the choice of model and experimental details, how to do this will change. I am not able to suggest options for this without understanding the experimental details in this setup.
- Addition of a related work section outlining and reviewing existing work and literature in the field, relevant to this work.

**Strengths And Weaknesses:**

- A lack of thorough review of existing work/literature within the domain of this paper is missing. This would make the paper hard to position relative to the state of the field if the audience does not have domain expertise.
- Details regarding the experimental setup are missing:
    - For the images in Figures, is this conditional or unconditional generation? If Conditional, how was conditional generation achieved? If a text-to-image model was used, what is (are) the prompt(s) used to generate the images?
    - What is the diffusion model used in this study? Did the authors train a new model from scratch? Or did the authors use an existing pre-trained model?
    - How many inference steps were used for generation? What are the values (or the range for) for the guidance parameters $\gamma_{cfg}, \gamma_{seg}$ used in this work?
- Some of the claims of this paper are not supported with sufficient empirical evidence.
    - "However, our experiments reveal that applying Gaussian blur only during the first 30–35% of the reverse diffusion steps is sufficient to guide the model toward stable solutions." This claim is not clear from Figure 2, which is used to support this claim.
    - Some of the claims from the results in section 3.1 are qualitative and not supported by sufficient (quantitative) evidence. Although at times image quality can be assessed visually, in most cases the differences are hard to determine by looking at the images. To name some examples:
        - "Figure 4 demonstrates that the optimal range for β in the EMA technique is close to 1, emphasizing the need for strong smoothing." This claim is backed only by qualitative evidence (looking at the images), there is no quantitative support from Fig 4 or other results.
        - Likewise for the claim "When applying high-intensity blurring with β nearly equal to 1, the final generated image exhibits significant improvements compared to those produced with lower β values or without energy guidance". It is hard to gauge whether the improvement is "*significant*" just form looking at the images.

---

> ### Author Response · Authors · 2025-05-04
> **Incorporated related work, more seeds & FID scores.**
>
> We have incorporated several key updates in this revised submission to address the concerns raised in the previous review. A comprehensive table outlining the various hyperparameters used in the energy-based guidance framework has been added to improve clarity. For better qualitative analysis, we now include results with three different random seeds in the main body of the text, enhancing readability and comparison. On the quantitative side, we have integrated Fréchet Inception Distance (FID) scores for the different smoothening techniques to facilitate a more rigorous evaluation. Importantly, we have also included a dedicated literature review section in this version, which was regrettably missing from our earlier submission. We sincerely apologize for this oversight. We hope these improvements comprehensively address the major concerns raised and enhance the overall quality and clarity of our work.

---

### Review · Reviewer_cWqf · 2025-04-05

**Summary Of Contributions:**

This paper reproduces the Smoothed Energy Guidance (SEG) method from a NeurIPS 2024 submission and proposes improvements to the original approach. It includes an ablation study on the effect of kernel size, which was not explored in the original work. The paper also examines alternative blurring strategies, such as Exponential Moving Average (EMA) and BoxBlur using integral images, as replacements for Gaussian smoothing. In addition, it introduces a modified smoothing schedule that avoids applying smoothing at every denoising step, aiming to improve computational efficiency while maintaining image quality.

**Audience:**

Yes

**Broader Impact Concerns:**

None.

**Claims And Evidence:**

No

**Requested Changes:**

- I kindly suggest the authors to add more qualitative and quantitative evidence, including more image generation metrics.

**Strengths And Weaknesses:**

**Strengths**

- This paper is clearly written and easy to follow.
- This paper the original method by enabling more flexible hyperparameter settings and enhancing computational efficiency.

**Weaknesses**

- The qualitative results are limited, and the ablation studies lack quantitative evidence. The comparison between the original method and EMA/BoxBlur only includes a time complexity analysis.
- The original SEG paper evaluates image generation quality using metrics such as FID. This paper does not report any metrics to assess the quality of the generated images.
- The contributions appear limited, as the paper mainly conducts additional ablation studies and evaluates standard techniques like EMA. Given that the original paper provides usable code, it is unclear whether the scope and novelty meet the expectations of the challenge described at https://reproml.org/call_for_papers/.

---

> ### Author Response · Authors · 2025-05-04
> **Evaluated FID score for quantitative comparison**
>
> We have incorporated more seeds (=3) for better qualitative analysis of different blurring techniques (Gaussian blur, EMA, Box Blur) under their respective params. For quantitative comparison, we have incorporated Fréchet Inception Distance (FID) scores for each technique and their variations.

---

### Decision · Action_Editor_r2pJ · 2025-05-21

**Recommendation:** Reject

**Comment:**

During the rebuttal, the authors have not fully addressed the reviewers' concerns.

One major concern is that the empirical evidence supporting the claims is limited. No quantitative evaluations (e.g., FID, CLIP score) are provided, and many conclusions are based only on a small number of qualitative examples. Important experimental details, such as the choice of diffusion model, task setup, and hyperparameter settings, are missing, making the work difficult to verify or reproduce. Furthermore, claims about computational savings are not backed by actual runtime or memory measurements. The paper also lacks a thorough discussion of related work and sufficient context around SEG and alternative guidance strategies.

Therefore, we recommend the authors carefully revise the paper according to the reviewers' suggestions and consider resubmission after revision.

**Audience:**

The topic of the paper is of sufficient interest to the TMLR community.

**Claims And Evidence:**

This paper reproduces the Smoothed Energy Guidance (SEG) method and proposes several improvements, such as alternative blurring strategies (EMA, BoxBlur) and a modified smoothing schedule. The paper is clearly written, and the reproduction effort is appreciated.

**Resubmission Of Major Revision:**

The authors may consider submitting a major revision at a later time.

---

> ### Author Response · Authors · 2025-05-23
> **Request for Clarification on Final Decision Despite Substantial Revisions**
>
> Dear Editors and Reviewers,
>
> Thank you for your time and the effort you’ve invested in reviewing our submission. I am writing in response to the final decision and accompanying comments, which I received on May 22, 2025.
>
> I would like to respectfully express my concern and confusion regarding the grounds for rejection, particularly the statement that "the authors have not fully addressed the reviewers' concerns." On May 4, 2025, I submitted a thoroughly revised version of the manuscript that directly incorporated feedback from all three reviewers. This included the following major updates:
>
> **Quantitative Evaluation:** We added FID scores for all proposed variations to support our claims with stronger empirical evidence.
>
> **Experimental Details:** We included detailed descriptions of the diffusion model used, task setup, hyperparameters, and implementation specifics to ensure reproducibility.
>
> **Computational Claims:** We provided runtime comparisons between SEG and our proposed variants to support our claims of improved efficiency.
>
> **Related Work:** We significantly expanded the related work section, offering better context for SEG and a comparison with alternative guidance strategies.
>
> ***Alongside the revised manuscript, I also uploaded a pointwise, detailed response addressing each reviewer’s comment. I took care to ensure that each concern was individually acknowledged and clearly responded to in the rebuttal document on OpenReview.***
>
> *However, I acknowledge that I did not explicitly outline these changes in the manuscript itself (e.g., through a changelog or marked-up version), and as someone new to the academic submission process, I now understand this may have contributed to the perception that the concerns were not adequately addressed.*
>
> Given this context, I kindly ask for clarification on whether the updated manuscript and detailed reviewer responses were fully reviewed prior to the final decision. If not, I would be sincerely grateful for any guidance on how to formally request reconsideration, or whether a future resubmission—accompanied by clearer documentation of changes—would be encouraged.
>
> Thank you once again for your time and dedication to maintaining a fair and rigorous review process.